# A Comparative Study of Factor Graph Optimization-Based and Extended Kalman Filter-Based PPP-B2b/INS Integrated Navigation

**Shiji Xin** [1,2], **Xiaoming Wang** [1,3,*] , **Jinglei Zhang** [1], **Kai Zhou** [1] and **Yufei Chen** [1]

1    Aerospace Information Research Institute, Chinese Academy of Sciences, Beijing 100094, China; xinshiji21@mails.ucas.ac.cn (S.X.); zhangjinglei@aircas.ac.cn (J.Z.); zhoukai@aircas.ac.cn (K.Z.); chenyufei20@mails.ucas.ac.cn (Y.C.)
2    School of Electronic, Electrical and Communicating Engineering, University of Chinese Academy of Sciences, Beijing 100049, China
3    College of Resources and Environment, University of Chinese Academy of Sciences, Beijing 100049, China
*    Correspondence: wxm@aoe.ac.cn

**Abstract:** Recently, factor graph optimization (FGO)-based GNSS/INS integrated navigation has garnered widespread attention for its ability to provide more robust positioning performance in challenging environments like urban canyons, compared to traditional extended Kalman filter (EKF)-based methods. In existing GNSS/INS integrated navigation methods based on FGO, the primary approach involves combining single point positioning (SPP) or real-time kinematic (RTK) with INS by constructing factors between consecutive epochs to resist outliers and achieve robust positioning. However, the potential of a high-precision positioning system based on the FGO algorithm, combining INS and PPP-B2b and that does not rely on reference stations and network connections, has not been fully explored. In this study, we developed a loosely coupled PPP-B2b/INS model based on the EKF and FGO algorithms. Experiments in different urban road and overpass scenarios were conducted to investigate the positioning performance of the two different integration navigation algorithms using different degrades of inertial measurement units (IMUs). The results indicate that the FGO algorithm outperforms the EKF algorithm in terms of positioning with the combination of GNSS and different degrades of IMUs under various conditions. Compared to the EKF method, the application of the FGO algorithm leads to improvements in the positioning accuracy of approximately 15.8%~45.9% and 19%~41.3% in horizontal and vertical directions, respectively, for different experimental conditions. In scenarios with long and frequent signal obstructions, the advantages of the FGO algorithm become more evident, especially in the horizontal direction. An obvious improvement in positioning results is observed when the tactical-grade IMU is used instead of the microelectron-mechanical system (MEMS) IMU in the GNSS/INS combination, which is more evident for the FGO algorithm than for the EKF algorithm.

**Keywords:** PPP-B2b; inertial navigation systems; factor graph optimization; extended Kalman filter; integrated navigation

## 1. Introduction

The increasing demand for location services in emerging fields, such as autonomous driving and mobile robotics, has drawn much attention to the development of accurate, highly reliable, and continuously robust positioning technology. Global Navigation Satellite System (GNSS), one of the most important positioning technologies, can provide users with high-accuracy location, velocity, and time information continuously under all weather conditions [1,2]. PPP is a widely used real-time high-accuracy positioning method in combination with GNSS, which requires access to precise satellite orbit and clock products broadcasted through the internet in real time [3,4]. PPP-B2b services, as one of the main features of BDS-3, broadcast high-precision satellite clock and orbit correction information via geosynchronous earth orbit (GEO) satellites, aiming to provide real-time decimeter-level

positioning results to users [5–7]. Compared with the real−time service (RTS) provided via the internet, the PPP-B2b service broadcasts correction information through BDS GEO satellites, which allows the users to perform the PPP without any internet communication and greatly expands the application scenarios of PPP.

However, GNSS positioning performance can degrade in complicated urban environments due to factors such as signal blockage and multipath interference. To mitigate these problems and improve positioning performance in urban environments, one solution is to integrate GNSS with other sensors, such as inertial navigation system (INS), to compensate for the limitations of GNSS in the presence of signal dropouts or interference [8,9].

Many previous studies have demonstrated the efficient performance of integrated GNSS and INS system using either a loosely or tightly coupled model. Elsheikh et al. [10] showed that integrating the real-time single frequency PPP with a low-cost consumer-grade INS can provide a continuous and precise navigation solution with an horizontal sub-meter accuracy even when passing under bridges and overpasses, where the GNSS is unable to obtain a reliable solution. In this experiment, real-time clock and orbit products from National Centre for Space Studies (CNES) were used in the PPP process; thus, it required a stable internet access. Kan et al. [11] conducted a PPP/INS tight integration experiment in urban environments using SSR-corrected data from different analysis centers, and the results showed that the three-dimensional positioning error remained around one meter. In the aforementioned studies, real-time precise ephemeris sent via the internet from different institutions, such as CNES, were used for PPP/INS integrated navigation. In addition to the RTS service transmitted via the internet, the satellite-based PPP-B2b service provided an alternative way for real-time PPP application. Xu et al. [12] investigated the performance of PPP/INS loose integration using PPP-B2b corrections, and the results showed a positioning accuracy of 0.36 m in open environments and approximately 0.85 m in obstructed environments. In the abovementioned studies, the Kalman filter was adopted for the parameter estimation in the GNSS/INS integrated navigation due to its maturity and computational efficiency in implementation.

Factor graph optimization (FGO) [13,14] has been widely used in the field of robotic navigation and has demonstrated its superior accuracy and effectiveness over filter-based methods for solving Maximum A Posteriori (MAP) estimation problems, particularly in complicated environments. FGO achieves optimal state estimation by solving non-linear optimization problems [15]. These improvements benefit from FGO's iterative and temporal correction capabilities, as mentioned in [16]. Iterative linearization plays a significant role in minimizing the linearization errors in non-linear observation models. Additionally, the simultaneous utilization of all observations within the FGO window enhances the ability to withstand outliers. Therefore, FGO can be also used in various challenging GNSS scenarios, and T. Pfeifer et al. [17] demonstrate its strong potential in sensor fusion. Indelman et al. [18] show that utilizing FGO in GNSS/INS loosely coupled integration (LCI) results in superior performance compared to the extended Kalman filter (EKF) estimator when using simulated data. Wen et al. [19] evaluates the application of loosely coupled and tightly coupled algorithms using pseudoranges and INS measurements for real-time positioning, revealing that FGO outperforms the EKF estimator in the complex urban environments. Wen et al. [20] proposes a formula for single point positioning (SPP) and RTK positioning based on FGO and evaluates the algorithm's feasibility in the challenging urban environment of Hong Kong, showing significantly improved positioning accuracy compared to filter-based estimators. Zhang H. et al. [21] proposes a state estimation algorithm using FGO in continuous time for GNSS/INS navigation systems to address the problem of significant trajectory deviations when GNSS observations temporarily become unreliable, aiming to achieve smoother trajectory estimations for the system. Liu et al. [22] have proposed an invariant filter estimator for tightly fusing monocular/stereo visuals, IMU measurements, and pseudoranges from GNSS. Experiments have shown that this method provides a significant advantage in terms of computational load compared to FGO-based algorithms, while achieving comparable levels of accuracy. Wang et al. [23]

have proposed a factor graph optimization-based multi-GNSS real-time kinematic framework, in which continuously tracked double difference ambiguities are utilized to establish ambiguity constraints for position states that share common-view satellites within the window. Additionally, a marginalization-based carrier phase ambiguity propagation method is introduced to achieve more reliable and continuous ambiguity resolution. The results demonstrate that, in open urban environments, this method performs comparably to the traditional EKF-based RTK, achieving centimeter-level positioning accuracy. However, in complex urban environments, this method outperforms EKF, with a 69.6% improvement in 3D positioning accuracy.

However, current research on FGO mainly combines SPP or RTK with INS measurements, scarcely considering the use of PPP-B2b, which can provide high positioning accuracy without relying on base station and internet communication. Research on comparing PPP/INS integration using FGO and EKF has not been conducted in detail, especially for the combination of PPP-B2b and low-cost MEMS inertial navigation for navigation purposes.

## 2. Materials and Methods

The theory of PPP-B2b aided with low-cost inertial measurement unit (IMU) can be divided into four parts, including the recovery of PPP-B2b, PPP positioning based on BD3 B2b service, the models of PPP-B2b/INS LCI based on EKF, and the models of PPP-B2b/INS LCI based on FGO. These four parts are elaborated in the following subsections.

### 2.1. Recovering Precise Orbit Corrections, Clock Offsets, and DCBs with PPP-B2b

As it is known, it is highly complicated to access the precise satellite orbit and clock products for PPP processing. The orbit clock corrections in PPP-B2b message, broadcasted by BDS-3 GEO satellites, were used to correct precise satellite orbit corrections, clock offset corrections, and DCBs for BDS-3 and GPS in the satellite-fixed frame. It is essential to transform the satellite position corrections provided by the PPP-B2b message into the ECEF frame. The satellite position vectors $[\Delta O_x \quad \Delta O_y \quad \Delta O_z]$ can be calculated using Equation (1):

$$
\begin{cases}
e_r = \frac{r}{|r|} \\
e_c = \frac{r \times \dot{r}}{|r \times r|} \\
e_a = e_c \times e_r \\
\begin{bmatrix} \Delta O_x & \Delta O_y & \Delta O_z \end{bmatrix} = \begin{bmatrix} e_r \, e_a \, e_c \end{bmatrix} \cdot \begin{bmatrix} \Delta O_r \\ \Delta O_a \\ \Delta O_c \end{bmatrix}
\end{cases}
\tag{1}
$$

where $e_r$, $e_c$, and $e_a$ represent the components of the direction unit vector in the radial, along-track, and cross-track directions, respectively; $[\Delta O_r \, \Delta O_a \, \Delta O_c]^T$ is the orbit correction vector in the satellite-fixed frame. $r$ and $\dot{r}$ represent the satellite position and velocity vector of the broadcast ephemeris, respectively.

The precise satellite position vector $\begin{bmatrix} X & Y & Z \end{bmatrix}_{orb}^T$ can be calculated by applying Equation (2):

$$
\begin{bmatrix} X \\ Y \\ Z \end{bmatrix}_{orb} = \begin{bmatrix} X \\ Y \\ Z \end{bmatrix}_{brd} - \begin{bmatrix} \Delta O_r \\ \Delta O_a \\ \Delta O_c \end{bmatrix}
\tag{2}
$$

where $\begin{bmatrix} X & Y & Z \end{bmatrix}_{brd}^T$ represents the satellite position vector calculated based on the broadcast ephemeris. The satellite position vector $\begin{bmatrix} X & Y & Z \end{bmatrix}_{orb}^T$, calculated using Equation (2), is the satellite position with the antenna phase center (APC) as the reference point. In contrast, the orbit correction products provided by IGS ACs are with respect to the center of mass (CoM) of satellites. Therefore, the transformation between the CoM satellites and APC satellites should be taken into account. The relative correction can be derived from the latest "igs14.atx" file provided by IGS.

The correction of satellite clock offset is also provided by the PPP-B2b service. The precise clock offset can be calculated using the PPP-B2b clock correction parameter $C_0$ by applying Equation (3).

$$\overline{dt}^s_{B2b} = dt^s_{brd} - \frac{C_0}{C} \tag{3}$$

where $\overline{dt}^s_{B2b}$ represents the satellite clock offset derived from the broadcast ephemeris, $\overline{dt}^s_{B2b}$ is the precise orbit clock offset, and C denotes the speed of light.

It should be noted that the clock offsets of BDS-3 provided by PPP-B2b products are based on the B3I signal, which differ from the RTS clock corrections, and all are based on the ionosphere-free linear combination (IFLC). Therefore, the clock estimations of GPS without considering the hardware code bias during the estimation process caused the inaccuracy, which can be corrected using the corresponding differential code bias (DCB) parameters by applying Equation (4):

$$dt^C_{B2b,B3I} = dt^C - B^C_{s,B3I} \tag{4}$$

where $dt^C_{B2b,B3I}$ is the precise satellite clock offset at the frequency of B3I; $dt^C$ represents the actual satellite offset, and $B^C_{s,B3I}$ is the DCB between different systems and B3I.

### 2.2. Mathematical Model of Real-Time PPP-B2b

In order to eliminate the first-order ionosphere delay, it is usual to adopt the ionosphere-free (IF) code and phase combinations. The linearized IF model can be written as follows:

$$\begin{cases} P^G_{IF} = \alpha P^G_1 + (1-\alpha)P^G_2 = \rho + d\overline{t}^G_r - d\overline{t}^G_s + \delta T + \varepsilon_{P_{IF}} \\ L^G_{IF} = \alpha L^G_1 + (1-\alpha)L^G_2 = \rho + d\overline{t}^G_r - d\overline{t}^G_s + \delta T + \overline{N}_{IF} + \varepsilon_{L_{IF}} \\ P^C_{IF} = \alpha P^C_1 + (1-\alpha)P^C_2 = \rho + d\overline{t}^C_r - d\overline{t}^C_s + \delta T + \varphi \cdot DCB^C_{1,3} + \varepsilon_{P_{IF}} \\ L^C_{IF} = \alpha L^C_1 + (1-\alpha)L^C_2 = \rho + d\overline{t}^C_r - d\overline{t}^C_s + \delta T + \overline{N}_{IF} + \varepsilon_{L_{IF}} \end{cases} \tag{5}$$

where $P^G_{IF}$, $L^G_{IF}$, $P^C_{IF}$, $L^C_{IF}$ represent IF pseudoranges and carrier phases of GPS and BDS-3, respectively; $P^G_1 P^G_2 P^C_1 P^C_2$ represent pseudoranges of GPS and BD3 at different frequencies; and $L^G_1 L^G_2 L^C_1 L^C_2$ denote carrier phases of GPS and BD3 at different frequencies. $\alpha = f^2_i/(f^2_i - f^2_j)$ wherein $f_i, f_j$ are different frequency values; $\rho$ denotes the geometric distance from satellites and receiver; $d\overline{t}^G_r$, $d\overline{t}^C_r$ are the clock offsets of GPS and BD3 at the receiver, and $d\overline{t}^G_s, d\overline{t}^C_s$ are the clock offsets of GPS and BD3 at the satellites; $\delta T$ is the tropospheric delay, and $\lambda_{IF} \cdot \overline{N}_{IF} = \lambda_{IF}N_{IF} - b^s_{IF} + b_{r,IF} - B_{r,IF}$ is the ambiguity parameter absorbing several types of hardware biases. $DCB^C_{1,3} = B^C_{B1I} - B^C_{B3I}$ is the DCB for the frequency of $f_{B1I}.\varepsilon_{P_{IF}}$ and $\varepsilon_{L_{IF}}$ denote the unmodelled errors of ionospheric-free code and phase combinations.

When compared with RTS-PPP, the satellites orbit and clock errors are considered to be eliminated when the orbit and clock estimations are considered to be corrected by the PPP-B2b service. However, there are some occasions when the orbit and clock corrections cannot be matched because the orbit and clock corrections are asynchronous with the occasional update. It is fully recommended to extend the validity of the clock corrections from 12 s to 26 s, by using the final matched clock and orbit correction pair until they are synchronous with the update [24].

### 2.3. PPP-B2b/INS Loosely Coupled Integration Based on EKF

It is typical for inertial navigation with auxiliary information to apply EKF for GNSS/INS integrated navigation. Considering the ultra-low-cost IMU used in studies, this algorithm uses 15 dimensional error state vectors, including navigation state vectors and IMU error vectors, which can be expressed as follows:

$$\delta x = [(\delta r^n_{IMU})^T \quad (\delta v^n_{IMU})^T \quad \phi^T \quad \delta b^T_g \quad \delta b^T_a] \tag{6}$$

where $\delta r_{IMU}^n = \begin{bmatrix} \delta r_N & \delta r_U & \delta r_D \end{bmatrix}$ and $\delta v_{IMU}^n = \begin{bmatrix} \delta v_N & \delta v_U & \delta v_D \end{bmatrix}$ denote the position error vector and velocity error vector in the n-frame, respectively, and $\phi = \begin{bmatrix} \phi_N & \phi_E & \phi_D \end{bmatrix}$ is the three-dimensional attitude error vector; $b_g, b_a$ represent the errors of the gyroscope and accelerometer biases, respectively. After the error disturbance analysis, the error differential of attitude, velocity and position can be deduced and can be written as follows:

$$\dot{\phi} = -\omega_{in}^n \times \phi + \delta\omega_{in}^n - C_b^n \delta\omega_{ib}^b \tag{7}$$

$$\delta\dot{v}^n = \mathbf{C}_b^n \delta f^b + \mathbf{C}_b^n f^b \times \phi - (2\omega_{ie}^n + \omega_{en}^n) \times \delta\mathbf{v}^n + \mathbf{v}^n \times (2\delta\omega_{ie}^n + \delta\omega_{en}^n) + \delta\mathbf{g}_l^n \tag{8}$$

$$\delta\dot{r}^n = -\omega_{en}^n \times \delta\mathbf{r}^n + \delta\boldsymbol{\theta} \times \mathbf{v}^n + \delta\mathbf{v}^n \tag{9}$$

where $\omega_{ie}^n$ denotes the earth angular rotation rate vector in the $n$-frame; $\omega_{en}^n$ is the angular rate vector of the n-frame; $\omega_{in}^n = \omega_{ie}^n + \omega_{en}^n$ refers to angular velocity vector; $C_b^n$ is the coordinate transformation matrix from b-frame to n-frame; $\delta\omega_{ib}^b$ is the gyro measurement error; $f^b$ is the specific force vector in the b-frame and $\delta f^b$ is its error vector; $\delta\omega_{ie}^n$ and $\delta\omega_{en}^n$ denote errors of $\omega_{ie}^n$ and $\omega_{en}^n$, respectively. $\delta\mathbf{g}_l^n$ represents the local global error; $\delta\theta = \begin{bmatrix} \delta\lambda\cos\varphi & -\delta\varphi \cdot \delta\lambda\sin\varphi \end{bmatrix}$, where $\varphi$ refers to the latitude and $\delta\lambda$ and $\delta\varphi$ are errors of latitude and longitude; more details on the above three equations can be found in [8].

The gyro and accelerometer biases are modeled as first-order Gauss–Markov processes and can be expressed as follows:

$$\begin{aligned} \delta\dot{b}_g &= -\frac{1}{T_{gb}}\delta b_g + w_{gb} \\ \delta\dot{b}_a &= -\frac{1}{T_{ab}}\delta b_a + w_{ab} \end{aligned} \tag{10}$$

where $T_{gb}$, $T_{ab}$ are the correlations of the gyroscope and the accelerometer; $w_{gb}$ and $w_{ab}$ represent their corresponding driving white noise.

PPP-B2b/INS LCI establishes the system state differential equation according to the error equation of INS. The difference between the position and speed calculated by GNSS and INS mechanization is used as the measurement value, and the optimal solution of the carrier's speed, position, attitude, and INS sensor error is obtained with Kalman filtering. Meanwhile, the estimated result is inputted back to correct the INS. The typical state vector used in PPP-B2b/INS LCI can be expressed as Equation (6), and the state and observation equation for PPP-B2b/INS LCI can be written as follows:

$$X_{LCI,k} = \phi_{LCI,k,k-1}X_{LCI,k-1} + \Gamma_{LCI,k-1}\omega_{LCI,k-1}, \omega_{LCI,k-1} \sim (0, Q_{LCI,k}) \tag{11}$$

$$Z_{LCI,k} = H_{LCI,k}X_{LCI,k} + \varepsilon_{LCI,k}, \varepsilon_{LCI,k} \sim N(0, R_{LCI}) \tag{12}$$

$$Z_{LCI,k} = \begin{bmatrix} p_{GNSS,IF}^n \\ v_{GNSS,IF}^n \end{bmatrix} - \begin{bmatrix} p_{INS}^n \\ v_{INS}^n \end{bmatrix} \tag{13}$$

where $X_{LCI,k}$ and $X_{LCI,k-1}$ are the state vectors of LCI at epochs of $k$ and $k-1$, respectively; $\phi_{LCI,k,k-1}$ is the system transition matrix from epoch $k-1$ to epoch $k$; $\omega_{LCI,k-1}$ denotes vector of system noise at epoch $k-1$, and $\Gamma_{LCI,k-1}$ is the matrix of noise distribution; $Q_{LCI,k}$ represents the covariance matrix of state noise; $Z_{LCI,k}$ denotes the innovation vector of the difference of position $p_{GNSS,IF}^n$, $p_{INS}^n$ and velocity $v_{GNSS,IF}^n$, $v_{INS}^n$ between PPP-B2b position solution and INS solution of prediction. $\varepsilon_{LCI,k}$ denotes observation noise with prior covariance of $R_{LCI}$; $H_{LCI,k}$ is design matrix of LCI.

In order to use the basic equation of discrete-time Kalman filter conveniently, it is necessary to discretize Equation (11), including system transition matrix with discrete-time $\phi_{LCI,k,k-1}$ and driving white noise with equivalent discretization $\omega_{LCI,k-1}$.

However, PPP-B2b positioning solution is referred to a GNSS antenna phase center, while the positioning solution predicted with INS mechanization is based on the center of IMU, which is not geometrically aligned. Therefore, it is necessary to correct the corresponding lever arm offsets during the data fusion. The design matrix $H_{LCI,k}$ that considers the lever arm offset can be expressed as follows:

$$H_{LCI,k} = \begin{bmatrix} I_{3\times3} & 0_{3\times3} & \left(C_b^n l^b \times\right) & 0_{3\times3} & 0_{3\times3} \\ 0_{3\times3} & I_{3\times3} & H_\phi & 0_{3\times3} & -C_b^n\left(l^b \times\right) \end{bmatrix} \tag{14}$$

where $H_\phi = \left[\left(\omega_{ie}^n + \omega_{en}^n\right) \times C_b^n\left(l^b \times\right) + C_b^n\left(l^b \times \omega_{ib}^b\right) \times\right]$.

The algorithm structure of PPP-B2b/INS LCI based on the EKF is shown in Figure 1.

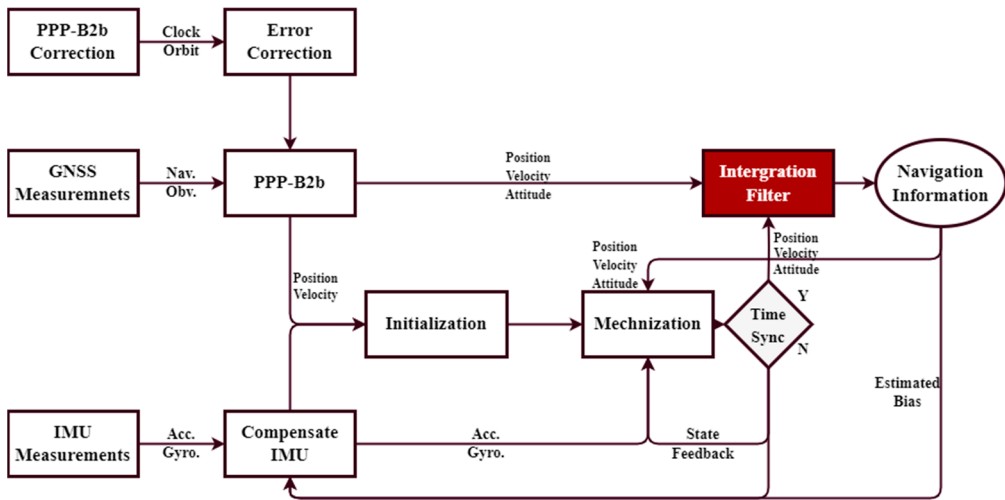

**Figure 1.** Flowchart of the system for PPP-B2b/INS integration based on the EKF.

*2.4. PPP-B2b/INS Loosely Coupled Integration Based on FGO*

In this section, we initially formulate and derive our state estimation problem within a probabilistic framework. The entire problem is structured as a factor graph, where the measurements from sensors form a series of factors that, in turn, constrain the system state. The three types of factors in the probabilistic graph will be discussed in detail in this section.

A.    Formulation

We define the optimal system state as the state that maximizes the posterior probability for all the provided measurements. Assuming all measurements are independent and the noise associated with each measurement follows a zero-mean Gaussian distribution, the Maximum A Posteriori (MAP) estimation problem can be further transformed into a minimizing sum of costs, where each cost corresponds to a specific measurement.

The optimization system, defined in this paper, aims to achieve the MAP estimation for all measurement values. It assumes that all measurements are independent and follow a zero-mean Gaussian distribution. The MAP problem is then transformed into the minimizing sum of costs, where each cost represents a measurement. For the FGO's PPP-B2b/INS LCI, the observations include the preintegration of IMU measurements and PPP-B2b positioning. To enhance the efficiency of the solver, a sliding window approach is adopted. The variables to be optimized within the sliding window $X = [x_0, x_1, x_2, \cdots, x_n]$ can be expressed as follows:

$$x_k = \left[\left(\mathbf{p}_{wb_k}^w\right)^T, \left(\mathbf{v}_{wb_k}^w\right)^T, \left(\mathbf{q}_{b_k}^w\right)^T, \left(\mathbf{b}_{g_k}\right)^T, \left(\mathbf{b}_{a_k}\right)^T\right]^T \tag{15}$$

where $x_k$ includes the position, velocity, and attitude of the world coordinate system at time $k$, as well as the biases of the gyroscope and the accelerometer. $n$ represents the size of the sliding window. Within each GNSS second, we introduce an IMU preintegration factor into the sliding window, regardless of the potential absence of GNSS positioning factors due to interruptions. Put differently, the preintegration factors are spaced at 1 s intervals. Additionally, the IMU preintegration factors are consecutive, indicating their temporal continuity. This aids in the estimation of IMU biases, as the biases are also included in the IMU preintegration factors. By minimizing the prior norm and the Mahalanobis norm of all measurement residuals, we arrive at the following MAP estimation:

$$
\min_{\mathbf{X}} \left\{ \left\| \mathbf{r}_p - \mathbf{H}_p \mathbf{X} \right\|^2 + \sum_{k \in [0,n]} \left\| \mathbf{r}_{Pre}\left( \hat{z}_{k,k+1}^{Pre}, \mathbf{X} \right) \right\|_{\Sigma_{k,k+1}^{pre}}^2 + \sum_{i \in [0,m]} \left\| \mathbf{r}_{PPP-B2b}\left( \hat{z}_i^{PPP-B2b}, \mathbf{X} \right) \right\|_{\Sigma_i^{PPP-B2b}}^2 \right\}
\tag{16}
$$

where $\mathbf{r}_{Pre}$ and $\mathbf{r}_{PPP-B2b}$ represent the residuals of the preintegration and PPP-B2b positioning observations, respectively. The next section will provide a more detailed description of the residual definitions. m represents the number of satellite positioning factors, and k represents the marginalized prior information, which will be discussed in detail later. The Ceres solver is used to handle non-linear optimization problems.

B.    IMU Preintegration Factor

The measurements involved in the inertial factors include biases, noise, linear accelerations, and angular velocities of the platform. Since accelerometers operate near the Earth's surface, the linear acceleration measurements also include the gravitational component. Taking into account the Coriolis and centrifugal forces produced by the Earth's rotation, the formula for the IMU in the EKF approach is modified accordingly. To better compare the performance of the EKF and FGO algorithms, a correction term for the Earth's rotation is introduced in the IMU preintegration [25].

In practical applications, the frequency of IMU measurements is typically an order of magnitude higher than the frequency of PPP-B2b positioning. Hence, individually estimating each state of the IMU measurements poses computational challenges. To address this issue, we employ the IMU preintegration method, which consolidates multiple measurements into a single measurement. The derived measurement for the inertial measurements within the time interval $[t_k, t_{k+1}]$ is calculated as follows:

$$
\begin{aligned}
\Delta \hat{p}_{b_{t_{k+1}}}^{b_{t_k}} &= \iint_{t \in [t_k, t_{k+1}]} \mathbf{R}_{b_t}^{b_{t_k}} \left( \tilde{\mathbf{a}}_t - \mathbf{b}_{a_t} \right) dt^2 \\
\Delta \hat{v}_{b_{t_{k+1}}}^{b_{t_k}} &= \int_{t \in [t_k, t_{k+1}]} \mathbf{R}_{b_t}^{b_{t_k}} \left( \tilde{\mathbf{a}}_t t - \mathbf{b}_{a_t} \right) dt \\
\hat{q}_{b_{t_{k+1}}}^{b_{t_k}} &= \int_{t \in [t_k, t_{k+1}]} \tfrac{1}{2} \mathbf{\Omega}\left( \tilde{\boldsymbol{\omega}}_t - \mathbf{b}_{w_t} \right) \hat{q}_{b_t}^{b_{t_k}} dt
\end{aligned}
\tag{17}
$$

With

$$
\mathbf{\Omega}(\boldsymbol{\omega}) = \begin{bmatrix} -\lfloor \boldsymbol{\omega}_\times \rfloor & \boldsymbol{\omega} \\ -\boldsymbol{\omega}^T & 0 \end{bmatrix}, \quad \lfloor \boldsymbol{\omega}_\times \rfloor = \begin{bmatrix} 0 & -\omega_z & \omega_y \\ \omega_z & 0 & -\omega_x \\ -\omega_y & \omega_x & 0 \end{bmatrix}
\tag{18}
$$

where $\tilde{\mathbf{a}}_t$ and $\tilde{\boldsymbol{\omega}}_t$, respectively, represent the measurements from the accelerometer and the gyroscope. $\mathbf{b}_{a_t}$ and $\mathbf{b}_{w_t}$ denote the biases of the accelerometer and the gyroscope. $\mathbf{b}_{t_k}$ represents the frame in the b system at time $t_k$, while $\left\{ \Delta \hat{p}_{b_{t_{k+1}}}^{b_{t_k}}, \Delta \hat{v}_{b_{t_{k+1}}}^{b_{t_k}}, \hat{q}_{b_{t_{k+1}}}^{b_{t_k}} \right\}$ represents the relative position, velocity, and attitude between b and a. It can be constructed without the initial position, velocity, and rotation of the IMU biases. Finally, the residuals associated with the system state and the preintegrated IMU measurements can be expressed as follows:

$$\mathbf{r}_{Pre}\left(\hat{z}_{k-1,k}^{Pre}, \mathbf{X}\right)$$

$$= \begin{bmatrix} \left(\mathbf{R}_{b_k}^w\right)^T \left(p_{wb_{k+1}}^w - p_{wb_k}^w - v_{wb_k}^w \Delta t_{k,k+1} - \frac{1}{2}g^w \Delta t_{k,k+1}^2 + \Delta p_{g/cor,k,k+1}^w\right) - \Delta \hat{p}_{b_{t_{k+1}}}^{b_{t_k}} \\ \left(\mathbf{R}_{b_k}^w\right)^T \left(v_{wb_{t_{k+1}}}^w - v_{wb_{t_k}}^w - g^w \Delta t_{k,k+1}\right) + \Delta v_{g/cor,k,k+1}^w \right] - \Delta \hat{v}_{b_{t_{k+1}}}^{b_{t_k}} \\ 2\left[\left(q_{b_{t_k}}^w\right)^{-1} \otimes q_{w_{i(k)}}^w(t_{k+1}) \otimes q_{b_{t_k}}^w \otimes \hat{q}_{b_{t_{k+1}}}^{b_{t_k}}\right]_{xyz} \\ b_{g_{t_{k+1}}} - b_{g_{t_k}} \\ b_{a_{t_{k+1}}} - b_{a_{t_k}} \end{bmatrix} \quad (19)$$

where $\mathbf{R}_{b_k}^w$ represents the transformation from the world coordinate system to the self coordinate system at time $k$, $g^w$ represents the gravitational acceleration in the world coordinate system, and $[\cdot]_{xyz}$ is the algorithm for extracting the quaternion (small-angle) rotation vector; $\Delta \mathbf{p}_{g/cor,k,k+1}^w$ and $\Delta \mathbf{v}_{g/cor,k,k+1}^w$ express the Coriolis correction components for the velocity and position preintegration, as previously defined. Additionally, the residuals encompass the online estimation and correction of gyroscope and accelerometer biases.

C.　PPP-B2b Positioning Factor

Generally, PPP-B2b positioning is based on the ECEF, ENU, and NED coordinates. Here, we consider the ENU coordinate as an example. By setting the first PPP-B2b positioning as the origin point, the PPP-B2b positioning in the ENU world frame, $\hat{p}_{PPP-B2b}^w$, and its covariance in the ENU direction can be obtained using the GNSS receiver. Thus, the residual of PPP-B2b positioning factor can be derived as follows:

$$\mathbf{r}_{PPP-B2b}\left(\hat{z}_i^{PPP-B2b}, \mathbf{X}\right) = \mathbf{p}_{wb_k}^w + \mathbf{R}_{b_i}^w l^b - \hat{p}_{PPP-B2b}^w$$

where $l^b$ is the lever arm offset expressed in the b-frame. The covariance in the ENU direction is mainly determined by the quality of the PPP-B2b positioning measurement.

D.　Marginalization

We employ marginalization to constrain the computational complexity of the sliding window optimizer. When the count of IMU preintegration factors surpasses a predefined threshold (equivalent to the size of the sliding window), the earliest IMU state is marginalized. Furthermore, the GNSS positioning factors associated with the IMU preintegration and marginalized states are converted into prior factors. For further insights into marginalization in sliding window optimization, refer to [26]. In this section, we integrate the enhanced IMU preintegration model into the FGO-based GNSS/INS integrated navigation system. Analytical expressions are utilized to compute the residuals for both the IMU preintegration factors and PPP-B2b positioning factors. We employ marginalization as a strategy to decrease computational demands.

E.　System Overview

Figure 2 shows the system for PPP-B2b/INS integration based on the FGO. We collect IMU measurements at the input frequencies of 200 Hz for tactical-grade IMU and 100 Hz for MEMS IMU. These measurements are deterministically extended for state variables and related IMU factors on the graph at a rate of 1 Hz. GNSS observations are expected to be available at a rate of 1 Hz as well. We employ the PPS signal from the GNSS receiver to calculate the time delay of the pre-processed GNSS observations. The PPP-B2b correction data needed for GNSS pre-processing is received and processed by the receiver in real-time to generate precise ephemeris. Following each optimization, the estimated IMU biases and re-calibrated gravity information are incorporated back into the IMU preintegration. To provide navigation solutions at a high frequency, we generate a state for each new IMU measurement obtained for the two optimization processes, enabling high-frequency state estimation.

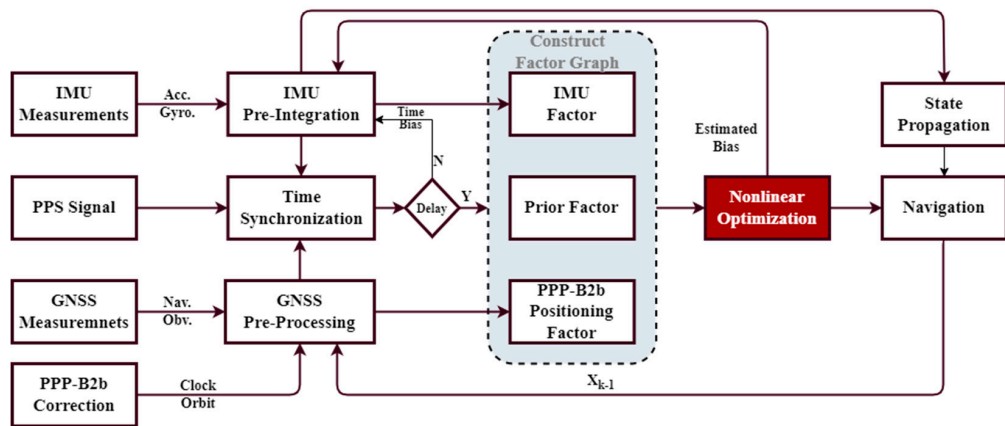

**Figure 2.** Flowchart of the system for PPP-B2b/INS integration based on the FGO.

### 3. Description of Experiments

In order to verify the performance of the proposed PPP-B2b/INS integrated navigation method based on FGO, two urban road vehicle experiments were conducted in Beijing, China. This article presents two experiments composed of several typical urban landscapes for detailed analysis. Experiment A took place from 371,514 to 374,911 s of 2269 week in GPS time. The red line represents vehicle trajectory, which is shown in Figure 3. The total length of the route was approximately 25.6 km. The experiment ran for 15 min in a relatively open environment, and then entered the downtown area with dense foliage, high-rise buildings, and overpasses. Experiment B took place from 375,476 to 377,401 s of 2269 week in GPS time. It was conducted in a more complex urban environment featuring tall buildings and significant tree obstacles.

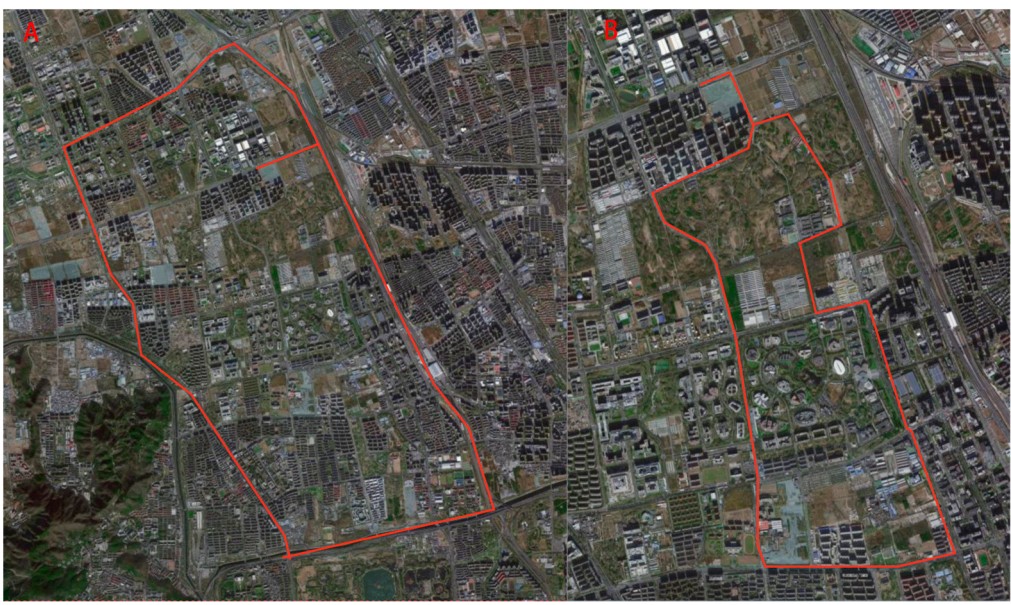

**Figure 3.** Trajectories of two vehicle experiments. (**A**,**B**) Experiment A conduct from 371,514 to 374,911 s of 2269 week in GPS time; Experiment B conduct from 375,476 to 377,401 s of 2269 week in GPS time.

As shown in Figure 4, the road vehicles were equipped with two types of receivers, the NovAtel PwrPak7 GNSS receiver and a receiver equipped with the ComNav W803 board, capable of real-time receiving and processing of PPP-B2b information. The raw observations of the GNSS were collected at a frequency of 1 Hz. At the same time, the road vehicles were equipped with two different IMUs, a tactical-grade IMU (ISA100C)

and a MEMS IMU (ADIS-16507). The detailed specifications about the IMU sensors are shown in Table 1. To ensure temporal consistency, hardware-level time synchronization was implemented to align the timestamps of various sensors to GPS time. For spatial synchronization, the offset between the IMU center and the GNSS antenna was accurately measured to calibrate the lever arm offset. The raw data from the tactical-grade and MEMS IMUs were recorded at frequencies of 200 Hz and 100 Hz, respectively.

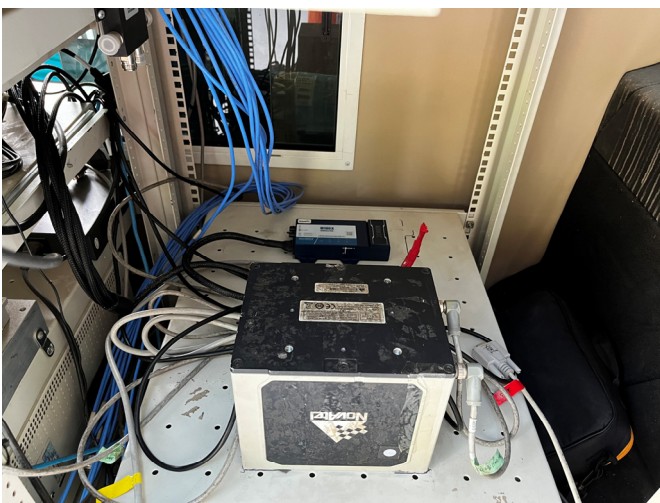

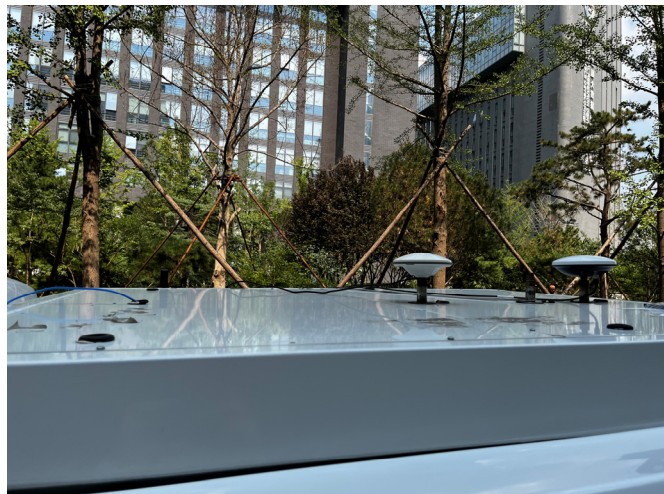

**Figure 4.** Illustration of the experimental equipment.

**Table 1.** Parameters of the two types of IMUs used in the experiments.

| IMU | Grade | Sampling | Bias | | Random Walk | |
| | | | Gyro (°/) | Acc. (mGal) | Angular (°/$\sqrt{h}$) | Velocity (m/s/$\sqrt{h}$) |
| --- | --- | --- | --- | --- | --- | --- |
| IAS100C | Tactical | 200 Hz | 0.5 | 100 | 0.03 | 0.1 |
| ADIS-15507 | MEMS | 100 Hz | 2.2 | 200 | 0.34 | 0.18 |

In both road vehicle experiments, a reference station equipped with the Septentrio PolaRx5 GNSS receiver was set up on the rooftop of Building D in the Aerospace Information Research Institute. This location provided a clear view of the sky. Using the raw observations from the tactical-grade IMU and two GNSS receivers, a smoothed solution based on tightly coupled multi-GNSS RTK and INS computation was obtained as the refer-

ence trajectory for the experiments. This computation was performed using the commercial software package IE 8.9.

In PPP-B2b positioning, this experiment used the B1C and B2a frequency bands of the BeiDou system, as well as the L1 and L2 signals from GPS satellites, for ionosphere-free (IF) combination. The PPP-B2b positioning results were obtained by processing the real-time received PPP-B2b signals. In the event of a PPP-B2b signal interruption, the orbit and clock corrections were performed by extending the PPP-B2b correction values [24]. If the interruption lasted for more than 10 min, the PPP-B2b correction values were discontinued. The more detailed PPP positioning strategies are shown in Table 2.

**Table 2.** Processing strategies for PPP-B2b.

| Item | Model |
| --- | --- |
| GNSS systems | GPS and BDS-3 |
| Elevation cut-off angle | 7 |
| Sampling rate | 1s |
| Phase wind-up effect | Model corrected |
| Ionospheric delay | Ionosphere-free linear combination with dual-frequency |
| Tropospheric delay | Dry component corrected by Saastamonien model; wet component estimated |
| Satellite antenna phase center | PCO and PCV values from igs14.atx |
| Receiver antenna phase center | PCO and PCV values from igs14.atx |
| Receiver clock | Epoch-wise estimated for each system |
| Ocean Tides | FES2004 |
| Phase ambiguities | Continuously static integer ambiguities are estimated |

## 4. Result and Discussion

In this section, the performance of PPP-B2b/INS integration navigation was studied in various environments, including open sky, urban canyons, and obstructed areas such as bridges and tree cover. The observations were processed using three modes: PPP-B2b, KF-based PPP-B2b/INS integration, and FGO-based PPP-B2b/INS integrated navigation. The performance of these three solutions was carefully analyzed and compared in terms of accuracy. The performance was analyzed by comparing the Root Mean Square Error (RMSE) of the positional differences relative to a reference benchmark to evaluate positioning accuracy.

### 4.1. Performance of PPP-B2b/INS Integration

Figure 5 shows the positioning sequences of Experiment A for PPP-B2b and EKF and FGO PPP-B2b/INS integration navigation. The figure also displays the number of available satellites (NSAT) and position dilution of precision (PDOP). The gray area in the diagram represents periods when satellite signals are interrupted or when the quality of satellite observations is poor. Within the initial 15 min after passing two bridges, the number of available GNSS satellites ranged from 11 to 14, and PDOP values were mostly below 2, ensuring continuous and reliable PPP positioning. However, as the vehicle entered a semi-urban operating environment with trees, high-rise buildings, and overpasses, signal tracking became intermittent, and NSAT frequently decreased. Due to frequent signal interruptions, the PPP float ambiguities converged multiple times, typically taking about twenty minutes to converge to a 4 decimeter-level accuracy. However, some outliers appeared in the PPP-B2b positioning sequence, attributed to ambiguity resolution failures. From Figure 5, it can be observed that the occurrence of outliers was accompanied by a decrease in NSAT, an increase in PDOP, or severe multipath errors. Furthermore, all these factors would reduce the accuracy of float ambiguities.

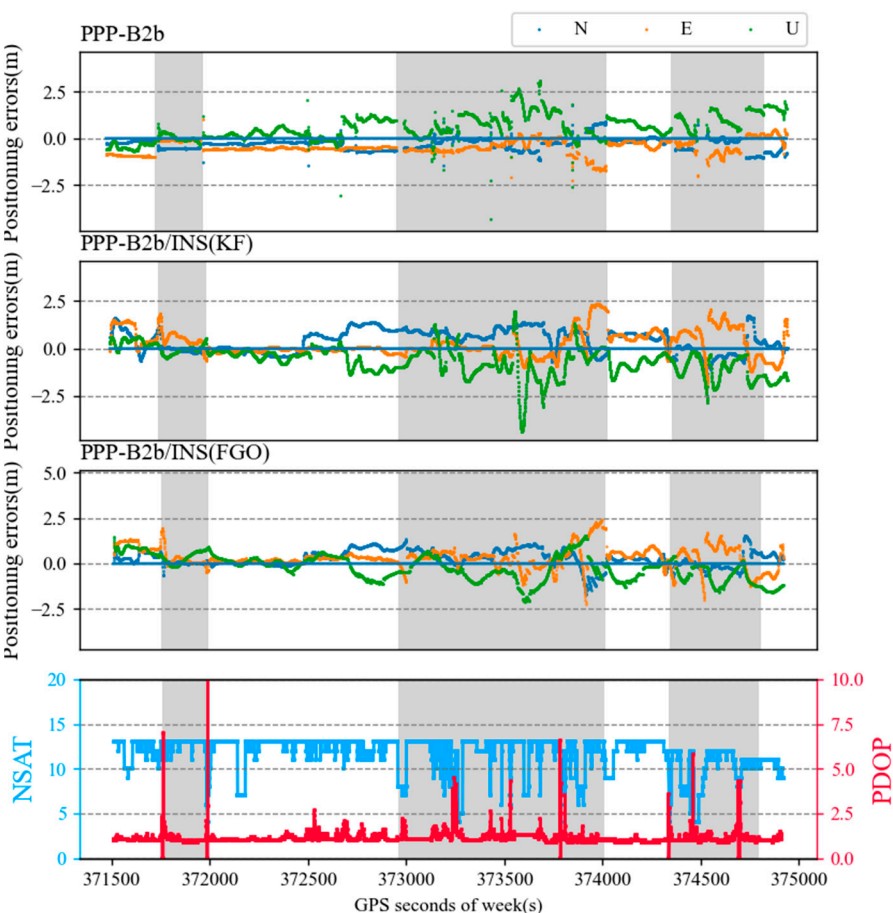

**Figure 5.** Positioning results series of PPP–B2b and EKF and FGO PPP–B2b/INS integrated navigation are, respectively, presented (Experiment A). The NSAT and PDOP series are also presented in the figure.

Figure 6 presents the specific observation environments of three scenarios, labeled as I, II, and III. The first scenario is a road transitioning from an overpass to an open-sky environment. The second scenario illustrates the vehicle crossing multiple overpasses in succession, during which GNSS signals experience frequent and brief interruptions. The third scenario depicts driving under an overpass, during which GNSS signal experienced a relatively long interruption. In a real-world kinematic environment, obstacles such as billboards, trees, and buildings inevitably lead to a decrease in the number of satellites and the GNSS measurement quality, thereby affecting ambiguity resolution. In the EKF-based PPP-B2b/INS integration, the utilization of high-frequency IMU data for state propagation can improve the continuity and smoothness positioning results. As indicated by the shaded area in Figure 7, from 372,961 to 374,011 s of week in GPS time, frequent signal interruptions occurred during a 1050 s driving period, with the longest interruption lasting 44 s. The PPP-B2b solution was almost unavailable during this period. Even with the data gap in GNSS, the EKF-based PPP-B2b/INS integration can provide continuous position estimates. However, the performance of the filter-based approach is closely related to the quality of the GNSS positioning results at the current epoch. Therefore, when the quality of PPP-B2b positioning is degraded, the system's performance will be significantly affected. In contrast, the FGO-based PPP-B2b/INS integration utilizes both the PPP-B2b positioning results at current and past epochs as well as the pre-integration information from IMU to obtain locally optimal navigation positioning information. As shown in Figure 5, the FGO method exhibits stronger robustness in segments with frequent signal interruptions compared to the EKF method, particularly in the U direction.

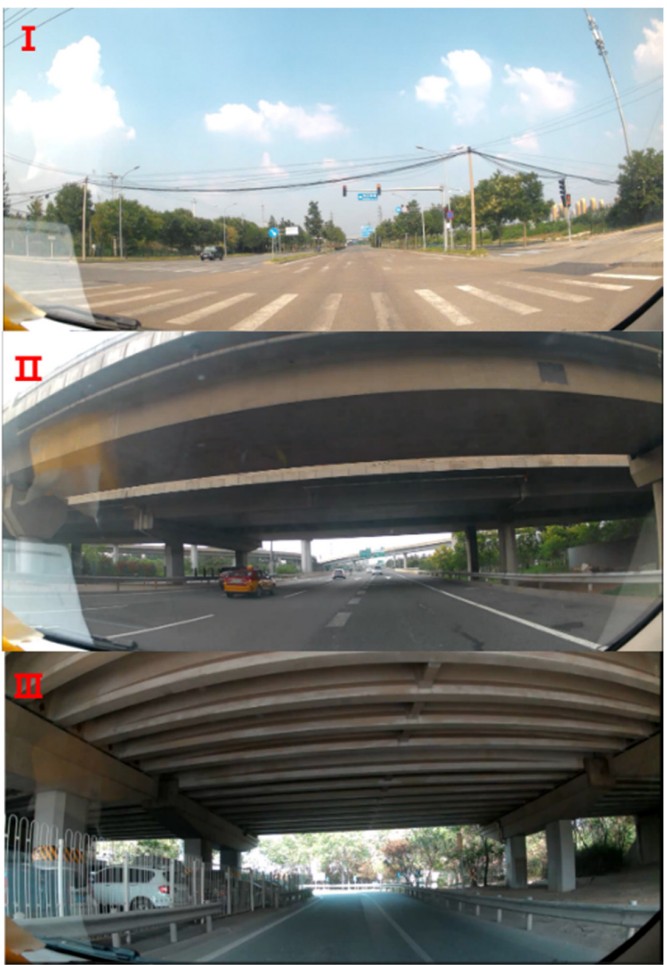

**Figure 6.** Three typical car driving scenarios in Experiment A (Number I–III represents three different observation environments).

To further compare the positioning performance of the two algorithms in complex urban environments, we selected three typical urban scenarios. The scenario I is a relatively open environment after passing through an overpass. The scenario II involves a city canyon with tree obstructions, leading to frequent short-term interruptions in GNSS signals. The scenario III is driving under a pedestrian bridge, causing longer periods of satellite signal loss. To quantify the positioning performance of both EKF and FGO algorithms, we collected data on the positioning errors over time throughout the entire driving period and during the three typical scenarios. Figure 7 shows the time-varying curves of horizontal error, zenith direction error, and 3D error for both algorithms during the entire driving process and the three typical scenarios. The red line represents the positioning error of the EKF algorithm, while the green line represents the positioning error of the FGO algorithm. From the graph, we can observe the following:

1.  In the scenario I, after the vehicle enters the relatively open area, the EKF algorithm achieves a stable horizontal accuracy of around 0.4 m for seven minutes, while the FGO method achieves a stable accuracy within 0.3 m, with it being 70% of the time within 0.2 m. Additionally, the vertical error fluctuates more significantly for the EKF algorithm compared to the FGO method.
2.  For the scenario II, characterized by frequent short-term interruptions due to city canyons and tree obstructions, the accuracy of positioning results obtained with both algorithms show a decrease compared to that in the scenario I. However, the FGO method exhibits better accuracy than the EKF method most of the time, and its performance is more stable.

3.  In the scenario III with longer signal interruptions under the pedestrian bridge, the accuracy of positioning results with these two methods shows a similar variation. However, the FGO algorithm consistently outperforms the EKF method, especially in the vertical direction.

Overall, the FGO algorithm demonstrates superior positioning performance compared to the EKF algorithm in the selected urban environments, especially in scenarios with signal interruptions and challenging environments.

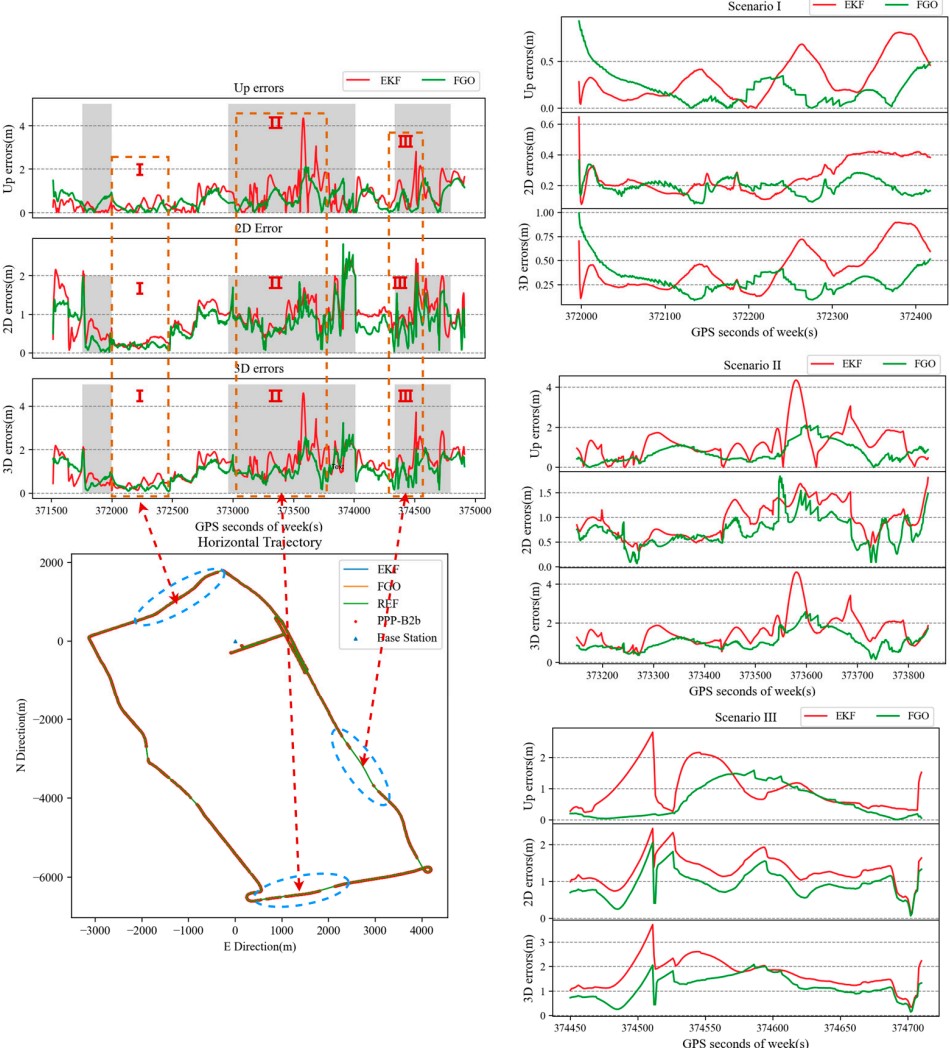

**Figure 7.** Positioning error series in three typical scenarios and the overall route, along with the driving trajectories. The trajectories for the three typical scenarios are marked on the trajectory plot.

Table 3 presents the RMSE of the positioning results obtained with FGO and EK in the three scenarios. In the horizontal direction, the RMSEs of the EKF algorithm for scenario I–III are 0.255 m, 0.951 m, and 1.314 m, respectively, with a mean RMSE of 0.916 for the entire period. For the FGO algorithm, the corresponding values are 0.734 m, 0.193 m, 0.727 m, and 0.932 m. In the vertical direction, the RMSE of the EKF algorithm for these three scenarios are 0.732 m, 0.317 m, 1.207 m, respectively, with a mean value of 1.314 m, while the FGO algorithm yields 0.586 m, 0.206 m, 0.743 m, and 0.932 m. Overall, the FGO algorithm exhibits a 19.85% improvement in the horizontal direction and a 19.92% improvement in the vertical direction over the EKF algorithm for the whole testing period. For the three scenarios in Figure 6, the FGO algorithm achieves improvements of 24.53%, 23.54%, and 29.064%, respectively, in the horizontal direction, and 34.98%, 38.42%, and 39.84%, respectively, in the vertical direction. From the data analysis, it is evident that

the FGO algorithm outperforms the EKF-based approach, particularly in scenarios with frequent signal interruptions, with a more pronounced advantage in the vertical direction.

**Table 3.** Position errors of PPP-B2b/INS of EKF and FGO and the improvement observed in Experiment A.

| Scene | Direction | EKF | FGO | Impro |
|---|---|---|---|---|
| **Scenario I** | Vert | 0.317 | 0.206 | 34.98% |
| | 2D | 0.255 | 0.193 | 24.53% |
| | 3D | 0.423 | 0.3 | 29.01% |
| **Scenario II** | Vert | 1.207 | 0.743 | 38.42% |
| | 2D | 0.951 | 0.727 | 23.54% |
| | 3D | 1.598 | 1.08 | 32.43% |
| **Scenario III** | Vert | 0.988 | 0.594 | 39.84% |
| | 2D | 1.314 | 0.932 | 29.064% |
| | 3D | 1.708 | 1.181 | 30.84% |
| **Total** | Vert | 0.732 | 0.586 | 19.92% |
| | 2D | 0.916 | 0.734 | 19.85% |
| | 3D | 1.253 | 0.996 | 20.55% |

### 4.2. Performance of PPP-B2b/MEMS Integration

The previous section's analysis of the PPP-B2b/INS integration algorithm was based on tactical-grade IMU. Compared to expensive high-performance inertial sensors, MEMS sensors exhibit the advantages of being lightweight, small in size, and cost-effective, making them more suitable for navigation applications such as autonomous driving vehicles, unmanned aircraft, and mobile robots. In order to further investigate the applicability of the FGO and EKF algorithms, Experiment B incorporated data from a low-cost MEMS IMU and was conducted in more challenging environments. The detailed parameters of the two IMUs used in Experiment B are shown in Table 1. As shown in Figure 8, Experiment B's travel route included two additional typical scenarios, a partially obstructed environment by trees and a heavily obstructed environment by trees, with the vehicle trajectory. The positioning errors of PPP-B2b, the tactical-grade PPP-B2b/INS (PPP-B2b/T-INS), and the MEMS-based PPP-B2b/INS (PPP-B2b/MEMS) are depicted in Figure 9. From the graph, it can be observed that PPP-B2b performs almost unusably in situations with poor observation quality, such as when there are few visible satellites or significantly elevated PDOP values. In such cases, PPP-B2b positioning requires assistance from INS to achieve continuous position estimation. The shaded areas in the graph represent challenging scenarios for both GNSS systems, where frequent signal interruptions occur and last for about 3 min. During this period, it can be seen that the position error of PPP-B2b/MEMS increases at a faster rate than the position drift of PPP-B2b/INS. This is because MEMS sensors typically exhibit a relatively poorer performance and stability compared to high-end inertial sensors due to their higher noise levels and significant bias instability.

To further compare the positioning performance of two integration navigation algorithms using different grades of IMUs in complex environments, we selected two typical scenarios in urban tree-lined road environments. The scenario IV involves one side of the road being obstructed by trees, resulting in a rapid decrease in the number of visible satellites and prolonged interruptions in the GNSS signal. The scenario V involves obstruction on both sides of the road, leading to frequent short-term signal interruptions and more prominent multipath effects. To quantify the performance of navigation algorithms using different grades of IMUs and different algorithms, we recorded the positional errors over time throughout the entire travel period and during the two typical scenarios. Figures 10 and 11 presents line plots of horizontal errors, zenith direction errors, and 3D errors over time for the entire travel process and during the two typical scenarios. The red lines represent the positioning errors using the EKF algorithm, while the green lines represent the errors using the FGO algorithm. From the graph, we can observe the following:

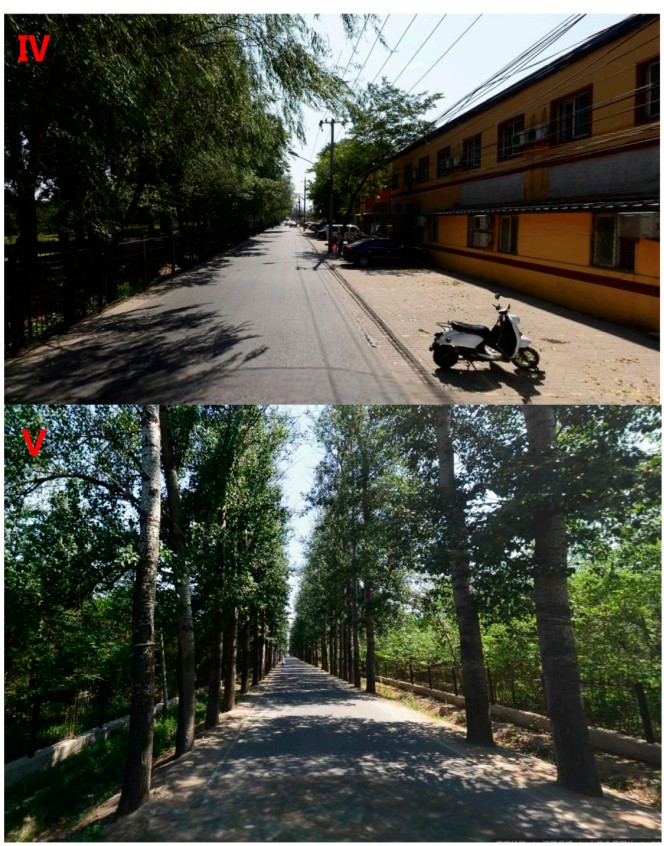

**Figure 8.** Another two typical car driving scenarios of Experiment B (Number IV and V represents two different challenging environments).

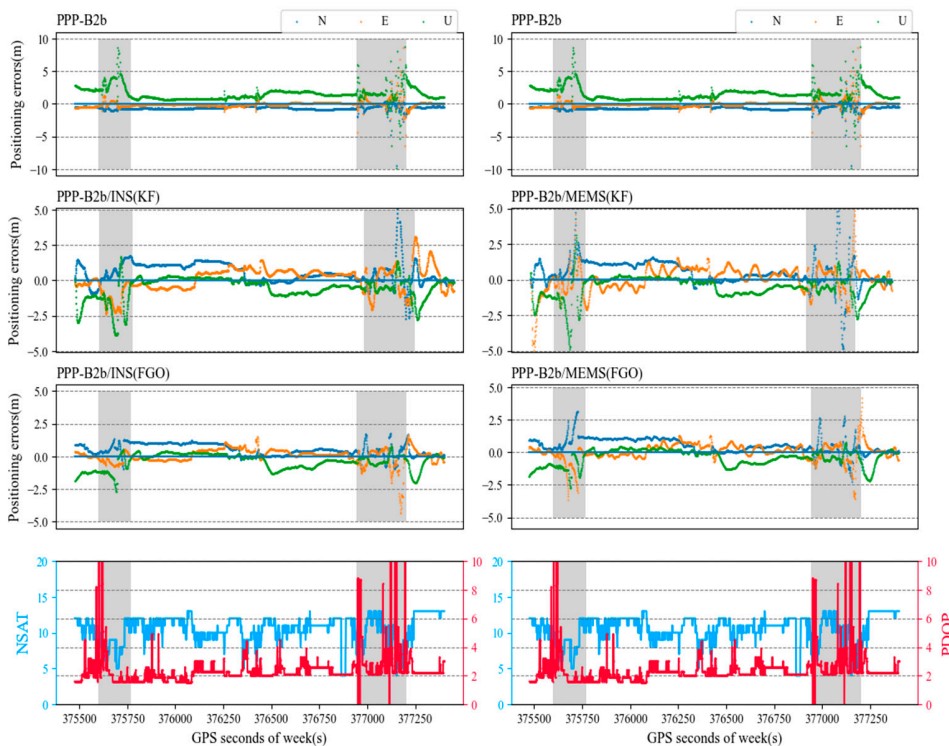

**Figure 9.** Positioning results series of PPP–B2b and EKF and FGO PPP–B2b/INS integrated navigation, respectively (Experiment B). The NSAT and PDOP series are also presented in the figure.

1. In Scenario IV, due to the obstruction caused by trees, the performance of combination navigation significantly deteriorates. The integration system becomes highly reliant on IMU predictions, causing the EKF-based PPP-B2b/MEMS to degrade into a mere inertial navigation system in the absence of satellite observations. Because of the high noise level of MEMS IMU, the errors quickly diverge. In contrast, the FGO algorithm can effectively suppress error divergence in this scenario. The navigation system using tactical-grade IMU performs better than MEMS IMU, with a slower error divergence, and the FGO-based integration navigation remains significantly superior to the EKF-based one.

2. In Scenario V, frequent short-term interruptions in satellite signals and a consistently low count of visible satellites result in significant degradation of PPP-B2b accuracy. In this situation, the EKF-based combination navigation heavily relies on INS predictions, leading to a relatively severe error divergence. The FGO algorithm, on the other hand, effectively utilizes information from multiple past epochs and uses higher quality positioning results within the sliding window as constraints. Regardless of whether tactical-grade IMU or MEMS IMU is used for the navigation system, the FGO algorithm outperforms the EKF algorithm in both horizontal and vertical directions, especially in scenarios with poor satellite observation quality, where the advantages of the FGO algorithm are more pronounced.

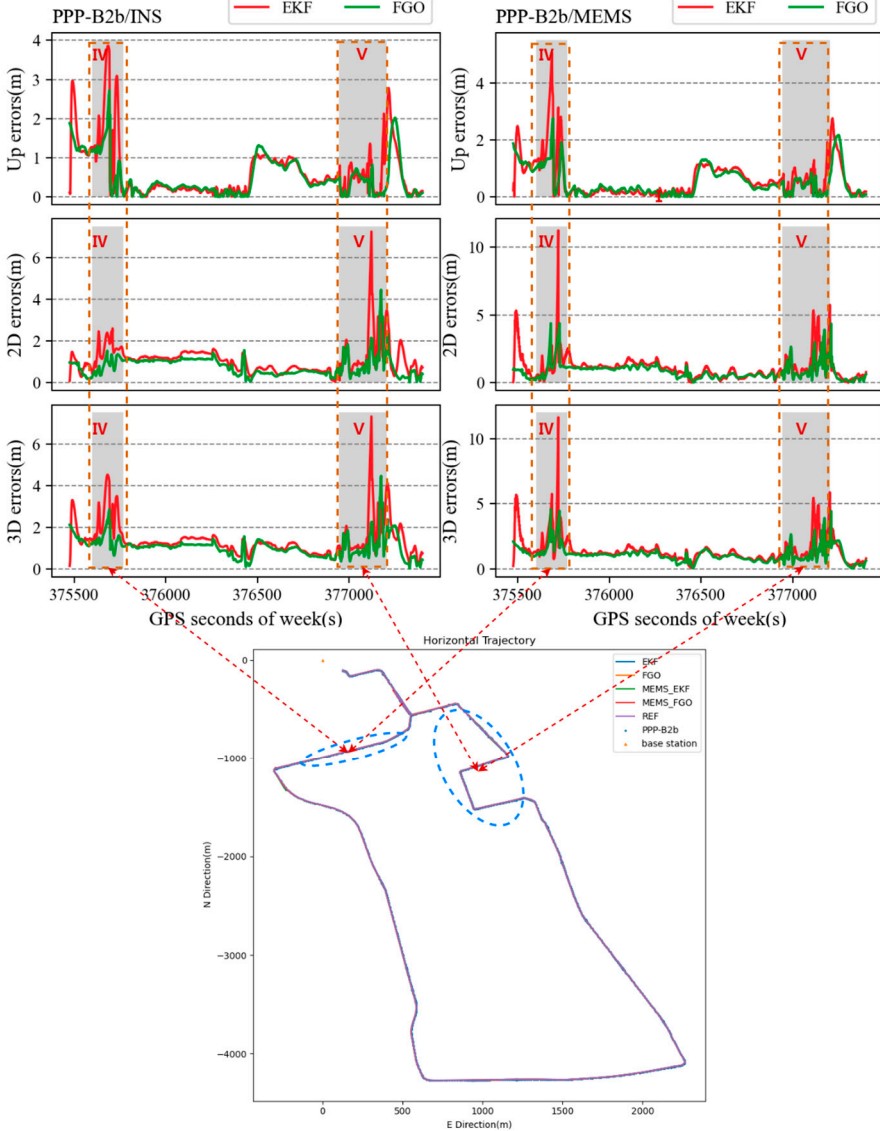

**Figure 10.** Positioning error series of the overall route, along with the driving trajectories. The trajectories for the two typical scenarios are marked on the trajectory plot.

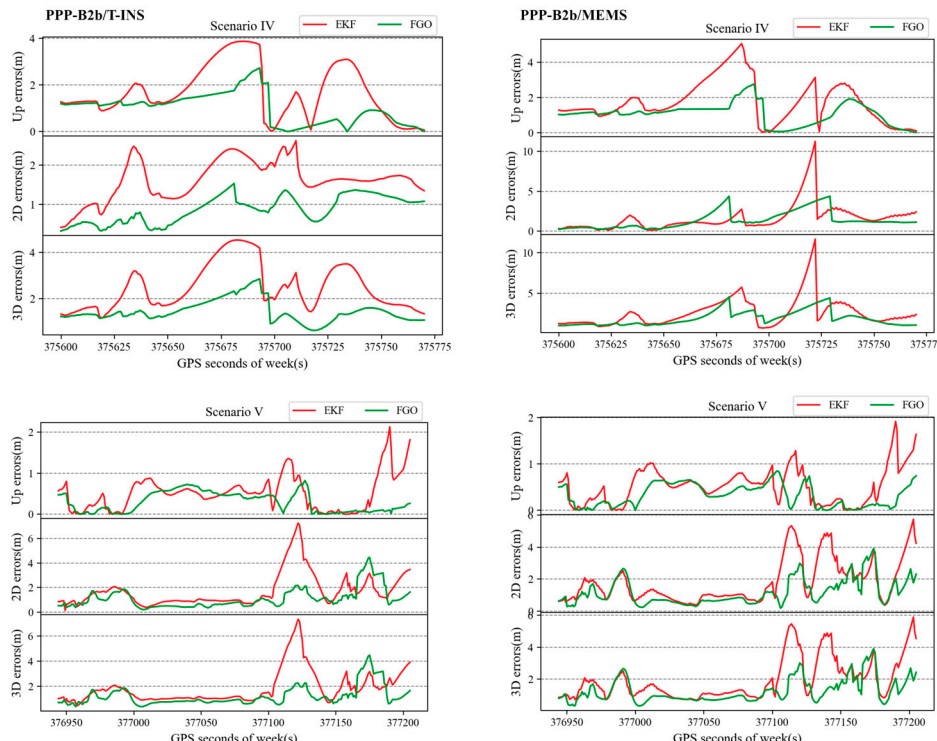

**Figure 11.** Positioning error series of PPP-B2b/T-INS and PPP-B2b/MEMS in two typical scenarios.

Overall, the FGO algorithm shows superior positioning performance compared to the EKF algorithm, particularly in challenging environments with degraded GNSS signals or limited satellite observations, and it consistently demonstrates advantages over the entire travel duration for both types of IMUs used.

Similarly, we calculated the positioning RMSE of PPP-B2b/INS using tactical-grade IMU and MEMS IMU, as shown in Tables 4 and 5. For PPP-B2b/T-INS integration navigation, the RMSE in the horizontal and vertical directions using the EKF algorithm for the first typical scenario were 1.626 m and 1.707 m, respectively. The FGO algorithm achieved 0.88 m and 1.003 m, representing improvements of 45.917% and 41.264% compared to the EKF algorithm, respectively. On the other hand, for PPP-B2b/MEMS combination navigation, the RMSE in the horizontal and vertical directions using the EKF algorithm for the first typical scenario were 1.629 m and 1.748 m, respectively. The FGO algorithm achieved 1.371 m and 1.071 m, representing improvements of 15.823% and 38.711% compared to the EKF algorithm, respectively. These results indicate that using tactical-grade IMU can significantly enhance the FGO-based integration navigation in specific complex environments compared to using MEMS IMUs, particularly in terms of horizontal accuracy. This is because the FGO algorithm utilizes the higher precision IMU pre-integration information to estimate the current state, resulting in better robustness against disturbances. Throughout the entire travel distance in Experiment B, the FGO algorithm using tactical-grade IMU achieved improvements of 18.849% and 29.494% in the vertical and horizontal directions compared to the EKF algorithm, respectively. When using MEMS IMU, the FGO algorithm achieved improvements of 19.084% and 20.079% in the vertical and horizontal directions, respectively. These findings also suggest that, in the FGO-based PPP-B2b/INS integration navigation, the IMU performance has a more pronounced impact on the system's horizontal performance in scenarios where there is a prolonged obstruction due to trees.

In addition, we have conducted statistics on the computational time cost of solving a floating-point PPP-B2b/INS solution for EKF and FGO in different experiments, as shown in Table 6. The average time required for the EKF algorithm to compute one epoch in Experiment A and Experiment B is 0.027 s and 0.029 s, with a maximum computational

cost of 0.053 s and 0.031 s, respectively. On the other hand, for the FGO algorithm, the average computation time in Experiment A and Experiment B is 0.098 s and 0.089 s, with a maximum computational cost of 0.522 s and 0.265 s, respectively. The computational cost of FGO is more than three times that of EKF, and at specific moments, FGO's computational cost exceeds half a second.

**Table 4.** Position errors of PPP-B2b/T-INS EKF and FGO and the improvement in Experiment B.

| Scene | Direction | EKF | FGO | Impro |
|---|---|---|---|---|
| **Scenario IV** | Vert | 1.707 | 1.003 | 41.264% |
| | 2D | 1.626 | 0.88 | 45.917% |
| | 3D | 2.474 | 1.45 | 41.379% |
| **Scenario V** | Vert | 0.514 | 0.295 | 42.725% |
| | 2D | 1.695 | 1.096 | 35.356% |
| | 3D | 1.837 | 1.196 | 34.904% |
| **Total** | Vert | 0.635 | 0.516 | 18.849% |
| | 2D | 1.115 | 0.786 | 29.494% |
| | 3D | 1.401 | 1.054 | 24.767% |

**Table 5.** Position errors of PPP-B2b/MEMS of EKF and FGO and the improvement in Experiment B.

| Scene | Direction | EKF | FGO | Impro |
|---|---|---|---|---|
| **Scenario IV** | Vert | 1.748 | 1.071 | 38.711% |
| | 2D | 1.629 | 1.371 | 15.823% |
| | 3D | 2.624 | 1.934 | 26.305% |
| **Scenario V** | Vert | 0.527 | 0.323 | 38.685% |
| | 2D | 1.807 | 1.169 | 35.28% |
| | 3D | 1.953 | 1.27 | 34.978% |
| **Total** | Vert | 0.639 | 0.517 | 19.084% |
| | 2D | 1.141 | 0.912 | 20.079% |
| | 3D | 1.444 | 1.166 | 19.241% |

**Table 6.** Statistics of the computational time cost of solving a floating-point PPP-B2b/INS solution for EKF and FGO in different experiments (Intel Core i7-12700H CPU).

| Modes | MEAN(s) | STD(s) | MAX(s) |
|---|---|---|---|
| EKF (Exp. A) | 0.027 | 0.009 | 0.053 |
| FGO (Exp. A) | 0.098 | 0.029 | 0.522 |
| EKF (Exp. B) | 0.029 | 0.002 | 0.031 |
| FGO (Exp. B) | 0.089 | 0.020 | 0.265 |

## 5. Discussion

In this study, we constructed a loosely coupled model of PPP-B2b/INS using both the EKF algorithm and the FGO algorithm, with the aim of exploring the positioning performance of these two different combination navigation methods in challenging signal environments. Additionally, we investigated and discussed the effects of different scenarios and INS types on precise positioning. To better understand the reasons behind the improved positioning accuracy of FGO compared to EKF, we conducted a statistical analysis of the positioning errors for FGO with sliding windows of 1 and 10, as well as EKF, as illustrated in Figure 12 and Table 7.

For PPP-B2b/T-INS integrated navigation, the FGO algorithm consistently demonstrates improvements in both horizontal and vertical accuracies compared to the EKF algorithm, especially in scenarios with signal obstructions or complex environments. These improvements mainly stem from two aspects.

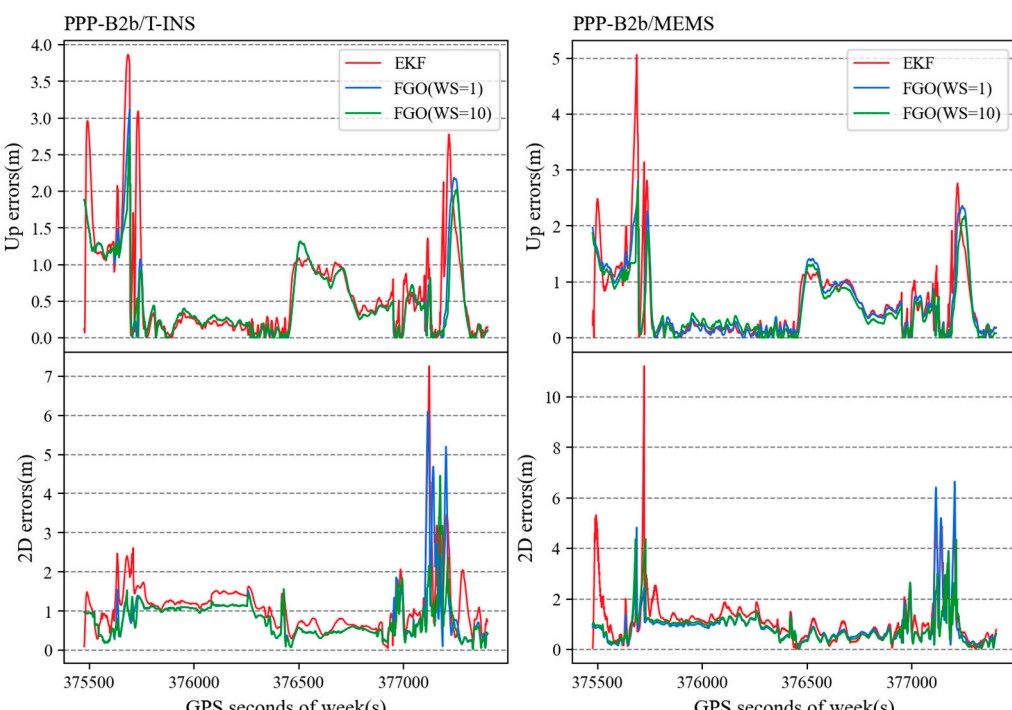

**Figure 12.** Positioning error series of EKF and FGO with window sizes of 1 and 10 in Experiment B.

**Table 7.** Position errors of EKF and FGO with window sizes of 1 and 10 in Experiment B.

|  | **Direction** | **EKF** | **FGO (WS = 1)** | **FGO (WS = 10)** |
|---|---|---|---|---|
| PPP-B2b/ T-INS | Vert | 0.635 | 0. 538 | 0.516 |
|  | 2D | 1.115 | 0. 866 | 0.786 |
|  | 3D | 1.401 | 1.150 | 1.054 |
| PPP-B2b/ MEMS | Vert | 0.639 | 0.584 | 0.517 |
|  | 2D | 1.141 | 0.900 | 0.912 |
|  | 3D | 1.444 | 1.210 | 1.166 |

Firstly, for PPP-B2b/INS loosely coupled integration, the EKF typically employs a single iteration to fuse PPP-B2b and INS data. In contrast, the FGO, as a process of determining the optimal estimation based on the gradient, involves multiple iterations within the sliding window and re-linearization, aiming to approach the optimal state [27]. This feature relaxes the requirement for the accuracy of initial estimator guesses and assists in handling non-linear situations. When the FGO window size is set to 1, meaning that the FGO method utilizes measurements from the previous epoch for optimal estimation, it behaves similar to the EKF estimator. In this case, the key difference between the FGO and EKF methods primarily lies in the number of iterations, as shown in Table 7. The FGO method with a window size of 1 (1.150 m and 1.210 m) outperforms the EKF method (1.401 m and 1.444 m) due to a higher number of iterations and re-linearizations, which is a key distinction between these two methods.

Secondly, the FGO method, compared to the EKF, takes into account more historical data rather than just information from a single epoch [19]. It optimizes the parameters of multiple epochs within the sliding window, effectively considering all historical information connected by INS factors. As seen in Figure 12, when satellite observation quality is good, the positioning performance of FGO with sliding windows of 1 and 10 is nearly identical. However, in adverse GNSS signal conditions, the error curve for FGO with a sliding window of 1 (blue) exhibits more drastic fluctuations compared to the error curve for FGO with a sliding window of 10 (green). Table 7 also indicates that the overall positioning

accuracy of FGO with a window size of 10 is better than that of FGO with a window size of 1. This is because the use of a sliding window enables the exploration and utilization of the time correlation between historical epochs, effectively resisting outliers and enhancing robustness towards outliers in GNSS measurements.

For PPP-B2b/MEMS integrated navigation, the FGO algorithm similarly exhibits consistent improvements in both horizontal and vertical accuracies compared to the EKF algorithm. A comparison reveals that FGO with tactical-grade IMU results in more significant improvements in horizontal positioning performance compared to with MEMS IMU. This is because higher grade IMUs provide more accurate preintegration information, leading to a higher quality of IMU preintegration information within the sliding window. This, in turn, facilitates resistance to outliers in PPP-B2b positioning solutions under adverse GNSS signal conditions.

In summary, as observed from Figures 7, 10 and 11, in an open-sky scenario (Scenario I), PPP-B2b/INS based on the FGO algorithm performs very similarly to the EKF method in terms of positioning accuracy, with horizontal errors at around 0.2 m. In such cases, EKF is evidently a preferable choice for PPP-B2b/INS integrated navigation due to its advantages in terms of efficiency and lower computational complexity. However, in relatively complex scenarios with features like trees, high-rise buildings, and overpasses (Scenarios II, III, IV, V), where signal tracking becomes intermittent, FGO exhibits noticeable improvements in the positioning performance. This is because, compared to EKF, FGO's advantages such as multiple iterations and time correlation make it a more recommended choice for PPP-B2b/INS integrated navigation in complex environments. However, one challenge faced in the application of FGO is that it requires the estimation of a larger set of parameters compared to the EKF estimator, which is referenced in Equations (6) and (15), and it involves repeated iterations and re-linearizations in each iteration, leading to an increased computational load. Table 6 shows that FGO consumes more time than EKF, even with the use of a sliding window. The computational efficiency of the EKF is significantly higher than that of the FGO algorithm, which could potentially limit the widespread application of FGO, especially in high-dynamic scenarios such as autonomous driving [15]. And this study is solely based on the loose coupling of GNSS and INS, with a relatively lower computational complexity. When adopting tight coupling or introducing other sensors such as vision and LiDAR, the FGO may exhibit a relatively greater difficulty than EKF in meeting the real-time navigation and positioning requirements of high-dynamic scenarios.

## 6. Conclusions

In this study, we conducted car experiments in different urban road and overpass scenarios to explore the positioning performance of two different integration navigation algorithms using different degrades of IMUs. The results indicate that the FGO algorithm outperforms the EKF algorithm in different environments and with different degrades of IMUs. The FGO algorithm improves the horizontal accuracy by approximately 15.8% to 45.9% and the vertical accuracy by 19% to 41.264%. In scenarios with prolonged signal obstructions, especially in environments with more challenging GNSS signals, the advantages of the FGO algorithm in horizontal positioning performance become particularly evident. Moreover, compared to using MEMS IMU, both the EKF and FGO algorithms with tactical-grade IMU exhibit better performances, and the FGO algorithm with tactical-grade IMU shows more significant improvements in the horizontal positioning performance. In conclusion, this study demonstrates the application advantage of the FGO algorithm in PPP-B2b/INS integration navigation, especially in complex environments and when using high-grade IMUs, offering further enhancements in positioning performance and providing a robust solution for scenarios with poor or interrupted GNSS signals.

However, the PPP-B2b/INS loosely coupled integration relies solely on the positioning results from GNSS without fully exploiting the raw GNSS measurements. This limitation can lead to severe degradation in the performance of the navigation system when poor PPP-B2b positioning results are used. In contrast, a tightly coupled approach directly

utilizes the raw GNSS observations, allowing for the detection and rejection of outliers at the raw measurement level [19,28]. Therefore, future research will focus on studying PPP-B2b in combination with IMU using tightly coupled methods based on both the EKF and FGO algorithms.

**Author Contributions:** Conceptualization, X.W. and S.X.; methodology, S.X.; software, S.X., J.Z. and K.Z.; investigation, S.X., J.Z. and Y.C.; data curation, S.X. and J.Z.; writing—original draft preparation, S.X.; writing—review and editing, S.X. and X.W.; supervision, X.W. All authors have read and agreed to the published version of the manuscript.

**Funding:** This research was supported by the funding program from the Aerospace Information Research Institute.

**Data Availability Statement:** The data sets generated and/or analyzed during the current study are available from the corresponding author on reasonable request.

**Acknowledgments:** The authors would like to acknowledge Xiaoji Niu and the Integrated and Intelligent Navigation (i2Nav) group from Wuhan University for providing the OB_GINS software that inspired the idea of this paper.

**Conflicts of Interest:** The authors declare no conflict of interest.

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
