# Peer review of "A Comparative Study of Factor Graph Optimization-Based and Extended Kalman Filter-Based PPP-B2b/INS Integrated Navigation"

_remotesensing, doi:10.3390/rs15215144_

Round 1
Reviewer 1 Report
In this paper, a loosely coupled PPP-B2b / INS model based on extended Kalman filter ( EKF ) and factor graph optimization ( FGO ) algorithm is proposed. Through experiments in different urban road and overpass scenarios, it is proved that the newly proposed model can still maintain high positioning accuracy and rich content in complex environments. However, the following problems need to be solved :
1. This paper lacks a comprehensive exposition of the application principle of the proposed PPP-B2b / INS model to improve the positioning accuracy in complex application scenarios, and shows how to solve the shortcomings of the EKF method to achieve the purpose of outstanding innovation.
2. In the experimental verification, a separate scene is used for verification. If you encounter more harsh environments such as multiple scene effects, does the proposed method still have good positioning performance?
3. Does this manuscript reference previous literature appropriately? If not, what references should be included or excluded?
4. The layout of some pictures needs to be adjusted so as to have a better connection with the corresponding picture description.
5. The experimental process in different experimental scenarios in the fifth part needs to be supplemented appropriately.
Minor editing of English language required.
Author Response
We are grateful for all the constructive and detailed comments from reviwer1. After conducting a reevaluation and reviewing the current literature on EKF and FGO, the manuscript has been revised based on all the suggestions from reviewers. Firstly, we have adjusted the proportion of different content in the introduction section, having reduced the introduction to GNSS and increased the research comparing FGO and EKF in the field of integrated navigation, corrected previous inaccurate statement. Additionally, we have added an analysis of why FGO provides better positioning accuracy than EKF in different scenarios and how higher-level inertial navigation can lead to more significant improvements, and have also highlighted some limitations of FGO in particular scenarios. Furthermore, we have improved the consistency in citation style, figures and tables placement, and overall manuscript organization and corrected the basic errors according to the reviews. More detailed revisions can be seen in the Word.

Reviewer 2 Report
The comparison of FGO and EKF algorithms is an interesting topic. In fact, some scholars have compared the performance of FGO and EKF algorithm in GNSS positioning, such as Weisong Wen and Xiaohong Zhang. The author conducted several tests, but did not analyze the essential difference between FGO and EKF. It is a great pity that the reader cannot know why the FGO algorithm is superior to EKF. The current manuscript is controversial. Detailed comments are as follows:
1. There seems to be no correspondence between the author and the work unit, please check.
2. There are redundant spaces in line 13.
3. FGO algorithm is a hot topic recently. As the title says, the focus of the article should be the algorithm performance comparison between EKF and FGO, and the reader is more concerned about this. However, the author does not explain the background and significance of comparing them in the abstract. In addition, it would be interesting if the author analyzed the reasons why FGO is better than EKF.
4, There is an unknown number in the keyword, please check the author!
5. The introduction section is too detailed about GNSS. Lack of comparison and analysis of EKF and FGO research status.
6. Line 46, blank space missing.
7, Lines 104-105, there are serious errors in the representation here, and I have reason to believe that the author did not conduct adequate background research, even if the references in the manuscript were not carefully read. At present, many researches have realized the combination of pseudo-range, Doppler and carrier phase at the observation level. In terms of positioning model, not only SPP but also PPP, RTK and PPP-RTK have been studied. In line 101, wen et al. [20], whose work is based on RTK, also studied the effect of ambiguity fixing. Therefore, the author needs to review the research status and the description of the contribution to this paper!
8. Line 110, missing spaces.
9. Line 159, variables of pseudoranges and carrier phases are incorrect..
10, Line 239, the text format is incorrect.
11. In Figure 1, the flow chart given by the manuscript should not only include FGO, but also EKF.
12. In Figure 2, the trajectory diagrams of the two vehicle-mounted experiments provide confusing content, and the author does not specify which colors are used to represent the trajectory. A clean and tidy background is recommended.
13, Lines 394-397, the reader does not find the multipath and noise errors mentioned by the author in Figure 5. Or the author's expression is not precise.
14. What does the gray rectangular area in Figure 5 represent and what is its significance? The author does not explain.
15. The curve in Figure 6 is clear, but the text in the subgraph on the right is very fuzzy. It is recommended that each subgraph be scaled at the same scale.
16. The author defines two scenarios in lines 491-492, but in Figure 8, the names of the scenarios are changed. The manuscript is confused.
17. There is no trace of the three scenes in Figure 9, please check. It is recommended that each subgraph be scaled at the same scale.
18. The discussion section was a bit boring, and the author just repeated the experimental results. The author needs to discuss the essential difference between FGO and EKF and the reason why FGO algorithm is better than EKF based on the experimental results.
19. Please note that the author needs to carefully check the manuscript before submitting the revision, too many basic errors will cause the reader to lose confidence for the manuscript.
Minor editing of English language required
Author Response
We are grateful for all the constructive and detailed comments from reviwer2. After conducting a reevaluation and reviewing the current literature on EKF and FGO, the manuscript has been revised based on all the suggestions from reviewers. Firstly, we have adjusted the proportion of different content in the introduction section, having reduced the introduction to GNSS and increased the research comparing FGO and EKF in the field of integrated navigation, corrected previous inaccurate statement. Additionally, we have added an analysis of why FGO provides better positioning accuracy than EKF in different scenarios and how higher-level inertial navigation can lead to more significant improvements, and have also highlighted some limitations of FGO in particular scenarios. Furthermore, we have improved the consistency in citation style, figures and tables placement, and overall manuscript organization and corrected the basic errors according to the reviews. More detailed revsions can be seen in the Words.

Reviewer 3 Report
The initial draft of the manuscript provides a reasonably well-structured background explanation of the technology under investigation in this research. However, I would like to highlight several important areas for improvement and further development.
Firstly, on line 104, there is a statement claiming that there is no existing research on the integration of FGO with current carrier-phase measurements and INS measurements. This assertion is not entirely accurate. A cursory keyword search on platforms like Google Scholar reveals the existence of several relevant papers on this very topic. Therefore, it is imperative that the authors undertake a more comprehensive review of these existing studies. Additionally, it is crucial to clearly articulate how this research distinguishes itself from the existing body of work.
Moreover, the paper primarily focuses on highlighting the advantages of FGO over the Extended Kalman Filter (EKF), specifically in terms of position accuracy. However, this comparison appears to be confined to specific conditions or scenarios. To bolster the paper's claims, a more profound and rigorous mathematical analysis is warranted. This analysis should not only demonstrate FGO's superiority but also elucidate the conditions under which it outperforms EKF and the potential limitations under different circumstances.
It is worth noting that comparing FGO with a loosely coupled EKF, rather than a tightly coupled EKF, raises questions about the applicability and generalizability of FGO's advantages. This choice of comparison should be explicitly justified and discussed in the paper.
Lastly, the manuscript's adherence to proper formatting and structure throughout the document needs substantial improvement. This includes consistency in citation style, figures and tables placement, and overall manuscript organization. Ensuring that the paper adheres to established publishing standards will enhance its readability and credibility.
In summary, while the initial draft provides a foundation for the research, it requires substantial expansion and refinement in various aspects. A more extensive review of existing literature, a deeper mathematical analysis, and improved formatting and structure are essential for the manuscript to make a meaningful contribution to the field of integrated navigation.
English is good to read and understand the manuscript
Author Response
We are grateful for all the constructive and detailed comments from reviwer3. After conducting a reevaluation and reviewing the current literature on EKF and FGO, the manuscript has been revised based on all the suggestions from reviewers. Firstly, we have adjusted the proportion of different content in the introduction section, having reduced the introduction to GNSS and increased the research comparing FGO and EKF in the field of integrated navigation, corrected previous inaccurate statement. Additionally, we have added an analysis of why FGO provides better positioning accuracy than EKF in different scenarios and how higher-level inertial navigation can lead to more significant improvements, and have also highlighted some limitations of FGO in particular scenarios. Furthermore, we have improved the consistency in citation style, figures and tables placement, and overall manuscript organization and corrected the basic errors according to the reviews. More detailed revisions can be seen in the Word.
